# Causality-Driven Feature Selection for Calibrating Low-Cost Airborne Particulate Sensors Using Machine Learning

**DOI:** 10.3390/s24227304

**Published:** 2024-11-15

**Authors:** Vinu Sooriyaarachchi, David J. Lary, Lakitha O. H. Wijeratne, John Waczak

**Affiliations:** Department of Physics, University of Texas at Dallas, Richardson, TX 75080, USA; vinu.sooriyaarachchi@utdallas.edu (V.S.); lhw150030@utdallas.edu (L.O.H.W.); john.waczak@utdallas.edu (J.W.)

**Keywords:** machine learning, causality, sensor calibration

## Abstract

With escalating global environmental challenges and worsening air quality, there is an urgent need for enhanced environmental monitoring capabilities. Low-cost sensor networks are emerging as a vital solution, enabling widespread and affordable deployment at fine spatial resolutions. In this context, machine learning for the calibration of low-cost sensors is particularly valuable. However, traditional machine learning models often lack interpretability and generalizability when applied to complex, dynamic environmental data. To address this, we propose a causal feature selection approach based on convergent cross mapping within the machine learning pipeline to build more robustly calibrated sensor networks. This approach is applied in the calibration of a low-cost optical particle counter OPC-N3, effectively reproducing the measurements of ${\mathrm{PM}}_{1}$ and ${\mathrm{PM}}_{2.5}$ as recorded by research-grade spectrometers. We evaluated the predictive performance and generalizability of these causally optimized models, observing improvements in both while reducing the number of input features, thus adhering to the Occam’s razor principle. For the ${\mathrm{PM}}_{1}$ calibration model, the proposed feature selection reduced the mean squared error on the test set by 43.2% compared to the model with all input features, while the SHAP value-based selection only achieved a reduction of 29.6%. Similarly, for the ${\mathrm{PM}}_{2.5}$ model, the proposed feature selection led to a 33.2% reduction in the mean squared error, outperforming the 30.2% reduction achieved by the SHAP value-based selection. By integrating sensors with advanced machine learning techniques, this approach advances urban air quality monitoring, fostering a deeper scientific understanding of microenvironments. Beyond the current test cases, this feature selection method holds potential for broader applications in other environmental monitoring applications, contributing to the development of interpretable and robust environmental models.

## 1. Introduction

The human quality of life is intricately intertwined with the physical environment surrounding us. We are now experiencing an era marked by unprecedented, widespread, and intense changes in the global environmental state [1]. Driven by human activities and population growth, significant global warming and consequent climate change are disrupting the usual balance of nature, posing a fundamental threat to various aspects of life. Human health is particularly affected, most significantly through its relationship with air quality. Global change and air quality are intertwined [2,3], and air quality is intricately related to human health. Air pollution is one of the greatest environmental risks to health. The World Health Organization (WHO) estimates that 4.2 million deaths annually can be attributed to air pollution [4]. Poor air quality due to pollutants such as ozone, particulate matter (PM), and sulfur dioxide can lead to a variety of health problems, including asthma, heart disease, and lung cancer. Associations have been reported between the concentrations of various air pollutants and diabetes, mumps, and epilepsy [5]. While the effects of climate change on air quality vary by region, many areas suffer a decline in air quality in parallel with global environmental change. Shifting weather patterns, including changes in temperature and precipitation, is expected to raise levels of PM. Robust evidence from models and observations shows that climate change is worsening ozone pollution. Climate change is expected to affect indoor heating and cooling demand due to temperature changes, altering fuel use, and consequently the composition of the emitted air pollutants. Evidence suggests that without additional air pollution controls, climate change will increase the risk of poor air quality in the US [6]. Therefore, the current state of global change and the concurrent exacerbation of air quality degradation emphasize the need for enhanced environmental monitoring capabilities at appropriate spatial scales.

The Internet of Things (IoT) has proved pivotal in this respect, enabling real-time data collection with high spatio-temporal resolution through networks of interconnected sensors. However, accessible, wide-scale deployment for environmental monitoring at more localized scales requires low-cost air quality sensor systems (LCSs) [7]. Although LCSs have the potential to bridge the gaps in sensor networks, thus facilitating dense monitoring networks capturing the relevant spatial variations in pollutants, they are less precise and have several sources of greater uncertainty compared to research-grade monitors. They are also more sensitive to environmental conditions. This makes them more likely to introduce potential measurement discrepancies compared to their reference sensors [8,9]. Therefore, LCSs require calibration prior to field deployment in order to improve the reliability and accuracy of the data being collected. This process involves the collocation of the LCS alongside a reference monitor at a representative location/s and then using the collected data to develop a calibration model that maps the raw output of the LCS to the measurements from the reference monitor. While several calibration mechanisms exist, machine learning is gaining popularity as a leading approach to LCS calibration [10,11,12].

### Machine Learning and the Need for Causality

Machine learning (ML) is a subset of artificial intelligence. It involves creating algorithms and statistical models that allow computers to learn from data and make predictions without the need for explicit programming. That is, learning through examples. ML has now gained immense popularity, and almost all sectors in the industrial arena leverage machine learning solutions to enhance productivity, decision-making processes, and other aspects [13,14,15,16].

It has proved useful in a wide variety of applications in science and engineering as well, especially for those applications where we do not have a complete theory, yet which are of significance. In the field of Atmospheric Science and Climate Physics, ML techniques are being used as an automated approach to build empirical models from data alone [17,18,19], and especially in air quality prediction [20,21,22,23].

Despite their wide usage and relative success, most traditional ML methodologies are constrained in their performance due to inherent limitations. One such limitation is the lack of interpretability [24,25,26]. Although adept at extracting patterns from data, ML systems develop complex models with numerous inputs that are often challenging to interpret. These models typically function as opaque black boxes, lacking the capability to elucidate the rationale behind their predictions or recommendations, or why a specific feature is prioritized compared to others in a model. Although model interpretation techniques such as SHAP values [27] are beneficial, they offer information solely on the functioning of the model that was learned, but not necessarily on how the variables under consideration relate to each other in the physical world. Empirical risk minimization commonly practiced in ML is designed to minimize a loss function on observed data, aimed at optimizing some performance metric. This approach is suboptimal since blindly optimizing for the performance on a finite dataset runs the risk of prioritizing associations within the dataset rather than actual cause and effect instances, thus leading to an incomplete problem formulation. In most scientific applications where we may seek to empirically model a phenomenon being studied, or where decisions are being made based on predictions from an ML model, and therefore unfavorable results have significant implications, interpretable ML models are essential. This is to foster not only scientific understanding and justification for model predictions, but also safety and reliability, since fully testing an end-to-end system, exhaustively anticipating every potential scenario where the system could fail, is not feasible for most complex tasks. In such cases, the incompleteness associated with the absence of sufficient interpretability can lead to unquantified bias [25]. This is one motivation for ensuring that feature–target pairs utilized by an ML model reflect genuine causal relationships in the real world and not mere statistical correlations present within a finite dataset. This closely ties to the other drawback, which is a lack of robustness or generalizability. That is the ability to be deployed in a different environment or domain than the one in which it was trained, and yield equivalent performance [28,29,30]. In supervised learning, the objective is to predict unknowns using available information by learning a mapping between a set of inputs and corresponding outputs. If due caution is not exercised, there is a risk of overfitting, where the algorithm fits too closely or even exactly to its training data. An overfit model would have learned and potentially prioritized the spurious correlations present within the dataset, specific to that particular distribution of data. Once the prediction environment diverges from the training environment, performance degradation should be anticipated since the model has learned to rely on superficial features that are no longer present. This is due to variations in the feature–target correlations between different environments. Determining whether the data generation process at the prediction time matches the training time is often uncertain, especially once deployed. In this vein, there are several studies in the literature that cite instances where ML models prioritize feature–target correlations specific to the datasets they have seen, and consequently generalize poorly to new unseen data [31,32,33]. ML models are by nature sensitive to spurious correlations [32]. Models relying on spurious correlations that do not always hold for minority groups can cause the model to fail when the data distribution is shifted in real-world applications. This is especially concerning for machine learning applications involving atmospheric and other environmental data, as is the case in LCS calibration for environmental monitoring, since the dynamic nature of the data renders it constantly shifting, sometimes even abruptly reaching extremes. This makes it virtually impossible to assert with certainty that the training data perfectly align with real-world instances once the model is deployed in the long term. Hence, in order to ensure that predictions made by ML models outside of the immediate domain of the training dataset remain accurate, we need models that are invariant to distributional shifts. That is, a model that would have learned a mapping which does not prioritize mere correlations, but rather features that affect the target in all environments. This need for invariance in ML models is another motivation for integrating causality into ML pipelines. This is because causal relationships are invariant. Mere correlation does not imply causation. If two variables are causally related, it should remain consistent across all environments. The invariance of causal variables is well established in the literature [34,35,36]. Consequently, a model making predictions exclusively using features directly causally related to the target should be more robust compared to general models.

Hence, in this study, we propose a feature selection step within our ML pipeline, based on causality, suitable for complex systems. The objective is to select a subset of relevant features from a larger set of available features based on a causal relationship to the target. Feature selection is a crucial step in developing machine learning models, aligning with the principle of Occam’s razor, which favors simpler hypotheses. Most conventional feature selection approaches employ filter methods where features are ranked or scored based on measures such as SHAP or LIME values [37], with a threshold applied to select the top ranked features. However, if the model has learned and relies on spurious correlations, the feature importance derived from these methods will also reflect those spurious relationships [30]. Consequently, relying on potentially misleading feature explanations compromises the generalizability we aim to achieve in our LCS calibration models. Another widely used approach is to rank features based on mutual information with the target variable. This is inadequate, as mutual information captures general statistical dependence rather than causal relationships. By adopting a causal approach to balance simplicity and predictive accuracy, we aim to leverage the well-established invariance of causal variables, enabling the development of more robust, accurate, and interpretable calibration models.

The study is structured as follows: In Section 2, we outline the proposed methodology for feature selection, detailing the underlying principles and techniques employed. This methodology is then applied to two case studies: the calibration of a low-cost optical particle counter OPC-N3 to reproduce ${\mathrm{PM}}_{1}$ and ${\mathrm{PM}}_{2.5}$ measurements from a research grade sensor. For comparison, we also develop calibration models for both case studies using two alternative approaches: (1) with all available features as predictors and (2) feature selection based on SHAP values. Section 3 presents the results, comparing the performance of the three approaches across the two case studies.

## 2. Materials and Methods

We first briefly outline the principle of convergent cross mapping (CCM), as developed in [38] and elaborated in [39], which forms the backbone of the proposed feature selection mechanism. CCM is a method that can distinguish causality from correlation that is based on nonlinear state-space reconstruction. This approach is specifically designed for nonlinear dynamics, which are predominant in nature and exhibit deterministic characteristics. Deterministic systems differ from stochastic ones primarily in terms of separability. In purely stochastic systems, the effects of causal variables are separable, meaning that information associated with a causal factor is unique to that variable and can be excluded by removing it from the model. This implies that stochastic systems can be understood in parts rather than as an integrated whole. This does not hold true for most complex dynamic systems such as climate systems, ecological systems, biological systems, etc., which do not satisfy separability. Therefore, CCM provides a more rigorous and overarching causal mechanism more suited for the complex dynamics of the datasets commonly dealt with in machine learning problems, especially in environmental sciences.

CCM is based on the principle that if a system possesses deterministic aspects and its dynamics are not entirely random, then there exists a structured manifold that governs these dynamics, exhibiting coherent trajectories. In dynamical systems theory, two time-series variables, *X* and *Y*, are causally linked if they are coupled and part of the same underlying dynamic system, which is represented by a shared attractor manifold, *M*. In such a case, each variable contains information about the other. If *X* influences *Y*, then *Y* holds information that can be used to recover the states of *X*. CCM measures causality by determining how well the historical states of *Y* can estimate the states of *X*, which would only be possible if *X* causally influences *Y*.

Takens’ theorem [40] provides the theoretical foundation for this approach, stating that the essential information of a multidimensional dynamic system is preserved in the time series of any single variable. Therefore, a time series of one variable can be used to reconstruct the state space of the system (e.g., Figure 1). When *X* causally influences *Y*, the dynamics of the system can be represented by the shadow manifolds $M_{X}$ and $M_{Y}$, constructed from the lagged values of *X* and *Y*, respectively. These shadow manifolds map onto each other since *X* and *Y* should belong to the same dynamic system. Nearby points on $M_{Y}$ should correspond to nearby points on $M_{X}$, indicating a causal relationship. If so, *Y* can be used to estimate *X*, and vice versa. The degree to which *Y* can be used to estimate *X* is quantified by the correlation coefficient ρ between the predicted and observed values of *X*, a process referred to as cross mapping. As the length of the time series increases, the shadow manifolds become denser, improving the precision of cross mapping, a phenomenon known as convergent cross mapping, which is the key criterion for establishing causality. The convergence property is crucial for distinguishing true causation from mere correlation. The degree to which the predictive skill converges can be interpreted as an estimate of the strength of the causal relationship.

As detailed in [38], CCM is distinct from other cross-prediction methods, as it focuses on estimating the states of one variable from another, rather than forecasting the future states of the system. This distinction is particularly important in systems with chaotic dynamics, where prediction can be hampered by phenomena such as Lyapunov divergence. CCM also handles non-dynamic, random variables, making it a robust tool for causality detection in complex systems.

### 2.1. Proposed Feature Selection Mechanism

We now elaborate our novel feature selection scheme. It is important to note that there may be other causally inspired feature selection methods in the literature. An example would be the automatic feature selection method for developing data-driven soft sensors in industrial processes proposed in [41]. That approach asserts that the capacity of a feature to reduce the uncertainty of a target variable, as measured by Shannon entropy, quantifies the causal impact of that feature on the target. Our approach is not intended to compete with such methods; rather, ours is designed specifically to handle the complexity of environmental and climate data. While information theory and entropy-based causal inference methods might be well suited for random variables, for atmospheric and environmental variables exhibiting complex interplay between various factors, CCM provides comparatively more solid criteria for causation, rigorously rooted in dynamical systems theory. The more generalized approach of CCM is more compatible with atmospheric and climate data that possess both stochastic as well as deterministic aspects, and due to its ability to identify weakly coupled interactions, which can play a significant role in complex systems where components influence each other but do not directly cause drastic changes or exhibit intricate feedback relations, we deem this a more suitable causal approach for the type of intricate systems addressed in environmental monitoring [39,42,43,44].

Hence, we propose a feature selection process for machine learning in which we leverage the principle of causation as imposed by CCM. Given a set of features {X} (in the case of LCS calibration, these would be individual output measurements from the LCSs along with external parameters such as ambient atmospheric pressure, temperature, and humidity to account for the sensitivity to environmental conditions) with the target variable *Y* (the target variable as measured by the reference instrument), our proposed work flow is as follows.

1:For each $X_{i}$ in {X}, the causation criteria set by CCM for $X_{i}\to Y$ is evaluated. For the current study, the causal-ccm package [45] was used for this purpose. The implementation details and steps of the CCM algorithm are described in Appendix A.In evaluating the causal relationship from $X_{i}$ to *Y*, it is essential to select a sufficiently long time series for both variables in order to ascertain that the criterion of convergence is met and that the cross-map skill does not deteriorate significantly over time.2:For each causality assessment, the causal-ccm package evaluates a *p*-value, representing the statistical significance of the result. All $X_{i}$ for which the *p*-value ≥0.05 [46] and therefore not registered as a sufficiently rigorous causal connection are eliminated from the set of input features to the ML model3:Next, the remaining features $\left\{X_{i}\right\}$ are ranked according to the strength of the causal relationship ρ, from most causally related to *Y* to the least.4:An appropriate threshold value is established for the strength of causality and the features exceeding this threshold are selected. The machine learning models are then constructed and trained for all possible subsets of the selected features as input variables to the model. After training, for each instance, the efficacy is tested using an independent validation dataset to assess how well it performs when presented with data that the algorithm has not previously seen, i.e., we test its generalizability.

By exploring various subsets of the most causally related features, as opposed to simply selecting the top-ranked ones, we aim to refine the selection process to retain to the most possible extent only the most direct causal influences. This approach seeks to enhance the generalizability of models by utilizing direct causal parents for predictions, as discussed in studies such as [35].

A reasonable choice of threshold for most cases would be ρ=0.5, since any feature with ρ≥0.5 retained for an appropriate duration of time would have established a causation guaranteed above chance and thus beyond being wholly attributed to noise, systematic error, or biases in the observational data. However, depending on the complexity of the system being modeled, the threshold may need to be adjusted to accommodate features with comparatively lower ρ values representing weak couplings that might offer important information to the model. Especially in climate systems, weakly coupled interactions are ubiquitous. An example of weakly coupled interactions can be found in the relationship between soil moisture and precipitation patterns. While soil moisture levels can influence local precipitation through mechanisms like evapotranspiration and land–atmosphere interactions, the coupling between soil moisture and precipitation is often not straightforward. However, understanding these weakly coupled interactions is crucial for accurate hydrological and climate modeling. By incorporating the nuanced effects of soil moisture on precipitation, models can better simulate regional water cycles, drought patterns, and the impacts of land surface changes on local climate conditions.

5:The model that demonstrates the best predictive performance is selected as the final calibration model. Performance metrics are compared with the full model to assess any improvement in generalizability. If no improvement is observed, the process in Step 4 is repeated using a lower threshold.

Figure 2 gives a concise representation of the proposed feature selection.

### 2.2. Experimental Test Cases

In order to validate the proposed feature selection method, it was applied on two real-world LCS calibration instances.

#### 2.2.1. Experimental Setup and Datasets Used

The two test instances were the calibration of a low-cost optical particle counter (OPC) to reproduce the ${\mathrm{PM}}_{1}$ and ${\mathrm{PM}}_{2.5}$ counts from a research-grade OPC.

The dataset was obtained from a previous study in [10]. The low-cost OPC used was a readily available but much less accurate Alpha Sense OPC-N3 (http://www.alphasense.com/) (accessed on 8 November 2024), together with a cheaper environmental sensor (Bosch BME280) (Bosch, Baden-Württemberg, Germany). The OPC-N3 adheres to the method defined by the European Standard EN 481 in calculating its PM values. A low-power micro-fan enables sufficient airflow through the sensor, at a sample flow rate of 280 mL/min. The OPC-N3, similar to conventional OPCs, measures the light scattered by individual particles passing through a laser beam in the sample air stream. Based on Mie scattering theory, the intensity of scattered light is used to determine the particle size and concentration. The OPC-N3 categorizes particles into 24 size bins that cover 0.35 to 40 μm, detecting nearly 100% of particles at 0.35 μm and around 50% at 0.3 μm, with a processing capacity of up to 10,000 particles per second. From these data, the mass concentrations of ${\mathrm{PM}}_{1}$, ${\mathrm{PM}}_{2.5}$ and ${\mathrm{PM}}_{10}$ are calculated, assuming particle density and refractive index. To convert each particle’s recorded optical size into mass, the OPC-N3 assumes an average refractive index at the laser wavelength (658 nm) of 1.5+i0. It is capable of on-board data logging and the on-board data are saved within an SD card which can be accessed through a micro-USB cable connected to the OPC [47].

The research-grade reference OPC used was a GRIMM Laser Aerosol Spectrometer and Dust Monitor Model 1.109 (Germany). It is capable of measuring particulates of diameters between 0.25 and 32 μm distributed within 32 size channels. The sensor operates at a flow rate of 1.21 L/min, and particulates entering the sensor are detected by scattering a 655 nm laser beam through a light trap. The scattered light pulse from each particle is counted, and the intensity of the scattered light signal is used to classify it into a specific particle size. A curved optical mirror, positioned at a 90° scattering angle, redirects scattered light to a photo sensor, with its wide angle (120°) enhancing light collection and reducing the minimum detectable particle size. The optical setup also optimizes the signal-to-noise ratio and compensates for Mie scattering undulations caused by monochromatic illumination, allowing a definite particle sizing [48,49].

The data were collected by collocating the LCSs and the reference sensor unit at a field calibration station in an ambient environment from 2 December 2019 to 4 October 2019. There were in total 42 initial input features to our ML model, which included the particle counts for each of the 24 size bins measured by the OPC-N3; the OPC-N3 estimates of ${\mathrm{PM}}_{1}$, ${\mathrm{PM}}_{2.5}$, and ${\mathrm{PM}}_{10}$; a collection of OPC performance metrics, including the reject ratio, in-chamber temperature and humidity; and the ambient atmospheric pressure, temperature, and humidity from the BME280. The target outputs for estimation were the ${\mathrm{PM}}_{1}$ and ${\mathrm{PM}}_{2.5}$ abundance as measured by the reference instrument, each with its own empirical model. The data were first resampled at a frequency of 60 s, and the different data sources merged by matching the time.

We resampled the data for several key reasons. First, to ensure an evenly sampled time series for causal analysis using CCM, which relies on time-delay embeddings. Second, since this study involved constructing and training several benchmark models for comparison, a compact dataset was necessary to minimize computational time. A resampling frequency of 60 s was chosen to create a compact dataset without undersampling, while still adequately capturing temporal variability in PM levels. Finally, instances with missing values (NaN) were dropped from the dataset.

The genre of ML used was multivariate nonlinear nonparametric regression. According to [10], the best-suited class of regression algorithms for the task is an ensemble of decision trees with hyperparameter optimization. Therefore, the GradientBoostingRegressor implementation of Python 3.10 was used for ML tasks. Of the final cleaned and data-matched dataset, 2130 data instances were isolated for hyperparameter optimization using cross-validation, a subset of which, a continuous time series of 600 time steps, was used for the causal feature ranking. The remaining dataset, consisting of 31,361 instances, was randomly partitioned into 80% for training and 20% for testing. We employed the train_test_split function from sklearn.model_selection with the shuffle parameter at its default value of True, for this purpose. To ensure the rigor of the process, there was no overlap between the training and testing datasets and the data used for causal analysis.

Separate calibration models were developed for each of ${\mathrm{PM}}_{1}$ and ${\mathrm{PM}}_{2.5}$. For each case, three approaches were employed and the results compared: (a) Using all 42 variables from the LCS as input features to the ML model; (b) using feature selection based on SHAP values; (c) using the proposed causality-based feature selection.

#### 2.2.2. ${\mathrm{PM}}_{1}$


First, all 42 output variables from the LCS described in Section 2.2.1 were used as input features with ${\mathrm{PM}}_{1}$ count from the reference sensor as the target variable for the ML model. The hyperparameters: the number of estimators (n_estimators), the learning rate (learning_rate), the maximum depth of the trees (max_depth), the minimum number of samples required to be at a leaf node (min_samples_leaf), and the number of features considered for splitting (max_features) were optimized using the GridSearchCV function of Python 3.10. To reduce the risk of overfitting, we constrained our grid search to smaller values of the learning rate (≤0.1) and a minimal range of 3–5 for the maximum depth of the trees. The optimized model was then trained on the training dataset and applied on the independent test dataset, and the performance of the model was assessed using the mean squared error (MSE) and the coefficient of determination (R²), two widely employed performance evaluation metrics in ML.

The SHAP values were then generated on the training dataset for the model to assess the model-specific contribution of each feature in predicting the ${\mathrm{PM}}_{1}$ count, and the features were ranked according to importance. A commonly used threshold for SHAP value-based feature selection is 0.5, indicating a significant influence on predictions. However, in our case, that would have eliminated most features (Figure 3) leading to underfitting. Therefore, for a fair comparison with the causality-based feature selection, the 10 highest-ranked features (highlighted in red in Figure 3) were chosen. ML models were then constructed with hyperparameter optimization and trained for all possible subsets of the selected features as inputs. Each instance was applied on the independent test dataset, and the performance metrics were generated. The best model was selected based on the MSE on the test set.

Next, the causality-based feature selection described in Section 2.1 was applied with threshold ρ≈0.7. A higher threshold of 0.7 was selected in this case because mapping LCS readings to reference-grade measurements is a relatively straightforward task, making it less likely that weakly coupled variables would have significant effects. Therefore, as an initial attempt, we used a threshold of 0.7 to include the top 10 highest-ranked features (Figure 4). Then, the best model was selected based on the MSE on the test set. For the time-delay embedding for CCM, since the time series were not overly sampled in time, the lag (τ) was set to one. The optimal embedding dimension (*E*) was empirically determined by applying simplex projection to the time series of the target ${\mathrm{PM}}_{1}$ counts from the reference sensor [44,50].

#### 2.2.3. ${\mathrm{PM}}_{2.5}$

The same procedure was followed for the estimation of ${\mathrm{PM}}_{2.5}$ particle counts, now with the ${\mathrm{PM}}_{2.5}$ count from the reference sensor as the target variable.

The feature importance ranking based on SHAP values is depicted in Figure 5. Since only two of the features placed above the threshold of 0.5, there also, the 10 highest-ranked features were considered for the subsequent feature refining.

The causality-based feature ranking is depicted in Figure 6, with threshold ρ≈0.51, that includes the top 10 ranked features.

## 3. Results

In this section, we present the results from each of the three approaches across the two test cases.

### 3.1. ${PM}_{1}$

Figure 7 presents a scatter diagram comparing the ${\mathrm{PM}}_{1}$ estimates from the OPC-N3 against those from the reference instrument prior to any calibration. As expected, the estimates from the OPC-N3 sensor showed substantial disparity compared to the reference instrument values, highlighting the need for calibration.

Table 1 depicts the performance evaluation metrics of the calibration model for ${\mathrm{PM}}_{1}$ derived from each approach, when applied on the independent test dataset. The causality-based feature selection was observed to yield the lowest MSE as well as the highest $R^{2}$ on the test dataset, demonstrating superior generalizability to unseen data and enhanced predictive performance. Therefore, the causality-based approach was clearly more adept at extracting the causal variables while eliminating the redundant, non-causal, and/or indirect influences on the target. It also used the least number of input features to the model out of the three. This improved computational efficiency, which is particularly valuable when sensors are deployed in the long term and at finer spatial resolutions in order to reduce the computational load of handling large datasets over extended periods.

Figure 8 shows the density plots of the residuals (that is, differences between the actual and predicted values) for each approach.

Figure 9 depicts the scatter diagrams of the calibration model for ${\mathrm{PM}}_{1}$ under different feature selection approaches. Figure 9a–c show the scatter plots of true vs. estimated ${\mathrm{PM}}_{1}$ count for each model on the training (blue) and independent test (red) sets. Figure 9d compares the performance of each model on the test set, with the causality-based approach yielding a comparatively thinner divergence from the 1:1 line. The density curve derived from the causality-based method exhibits a prominent density peak and narrower spread compared to the other two, indicating the most accurate predictions of the three, with fewer large residuals, thus producing a more reliable and robust model with fewer prediction errors.

### 3.2. PM_2.5_

Figure 10 depicts the scatter diagram comparing the ${\mathrm{PM}}_{2.5}$ estimates from the OPC-N3 against those from the reference instrument prior to calibration. Again, the necessity for calibration is underscored by the significant deviation in the estimates from the OPC-N3 sensor from the reference instrument values.

Table 2 presents the performance metrics of the ${\mathrm{PM}}_{2.5}$ calibration model derived from each approach evaluated on the independent test dataset.

Although MSEs were higher and the $R^{2}$ values were lower across all three approaches compared to ${\mathrm{PM}}_{1}$ calibration models, the causality-based feature selection method consistently yielded the lowest MSE and the highest $R^{2}$ by a reasonable margin, with the least number of input features to the model.

Figure 11 depicts the density plots of the residuals for the ${\mathrm{PM}}_{2.5}$ estimation. Both models incorporating feature selection exhibited improved accuracy compared to the model without feature selection. Although less pronounced than in the case of ${\mathrm{PM}}_{1}$ models, the causality-based model continued to exhibit the narrowest residual distribution over larger values, characterized by a smaller base spread and a slightly higher peak compared to the SHAP value-based approach.

Figure 12 gives the scatter diagrams of the calibration model for ${\mathrm{PM}}_{2.5}$ under different feature selection approaches, along with the comparison of the models’ performance on the test set. Again, though less pronounced than in the case of ${\mathrm{PM}}_{1}$ models, the causality-based approach yielded the thinnest divergence from the 1:1 line.

## 4. Discussion

Our results demonstrate the efficacy of the causality-based feature selection method in building more accurate and robust calibration models for LCSs that generalize better to unseen data.

We compared the performance of the proposed method with feature selection based on SHAP values, a common approach for machine learning practitioners [51]. The proposed causality-based feature selection method consistently outperformed the SHAP value-based approach. It is important to note that in an effort to provide a rigorous and thorough comparison with the proposed method, we opted for a minimal threshold (<0.2 in both test cases) for feature selection based on SHAP values. Therefore, the observed underperformance of the SHAP-based approach highlights its limitations in extracting causal information and reinforces the susceptibility of machine learning models to spurious correlations.

The proposed feature selection method was validated on the calibration of two OPC instances. This provided a compelling case study for assessing the novel approach, since the working principle of OPCs (see Section 2.2.1) involves the estimation of particle size distributions based on light scattering, and not only is the relationship between scattered light intensity and particle characteristics (such as size, shape, and refractive index) nonlinear, but particle behavior and optical properties can often be influenced by environmental conditions. These complexities provide a strong basis for utilizing CCM, which is specifically designed to uncover causal influences in complex, nonlinear systems with subtle dependencies. In both case studies, the features chosen as predictors from the proposed causal approach validated its ability to extract the most direct causal influences from mere correlations and indirect influences. In both instances, the reject count ratio was extracted as an important predictor. This can be attributed to the operational principles of the OPC-N3. The OPC-N3 comprises two photo diodes that record voltages which are subsequently translated into particle count data. Particles partially within the detection beam or passing near its edges are rejected, and that is reflected on the parameter “Reject count ratio”. Consequently, this parameter enhances particle sizing accuracy, hence having a direct influence on the PM count [10]. Several studies have recorded the impact of meteorological parameters such as atmospheric pressure, temperature, and humidity on atmospheric levels of PM [52,53,54,55,56,57,58]. Atmospheric pressure affects PM levels through its impact on air density, vertical mixing, and the transport and dispersion of particles. Atmospheric pressure obstructs the upward movement of particles. Under high-pressure systems, air tends to be more stable, trapping pollutants near the surface, increasing PM levels. In contrast, in lower-pressure conditions, particles may disperse more easily due to reduced air density. Temperature also affects atmospheric stability, changing how pollutants disperse. Hot weather often results in stagnant air conditions, which can trap PM and hinder its dispersion. In addition, elevated temperatures can speed up chemical reactions that generate PM, particularly in areas with vehicle and industrial emissions. Humidity influences ambient levels of PM significantly through hygroscopic growth: certain atmospheric species absorb water and increase in size once the relative humidity exceeds the deliquescence point of the substance. This phenomenon can shift smaller particles into larger PM-size categories, resulting in changed PM levels. High humidity can also promote chemical reactions, such as the conversion of sulfur dioxide to sulfate aerosols [59], leading to higher PM levels. In addition to the direct impact of meteorological factors on ambient PM levels, the performance of LCSs can also be influenced by these environmental conditions [60]. Therefore, naturally, ambient temperature, pressure, and humidity should be important predictors to the calibration model, which causality-based feature selection was able to extract, as opposed to the SHAP value-based approach, which placed greater import on temperature and humidity in the interior of the OPC leading to less accurate predictions on the test data. This also demonstrates the versatility and robustness of a causal feature selection method underpinned by CCM, as it precisely identifies the significance of causal relationships between sensor readings and ambient environmental variables consistently across both cases. Tangentially, as a control comparison to verify CCM’s superior compatibility, we implemented Granger causality [61], which failed to identify any statistically significant causal influence from ambient environmental factors on the PM abundance (see Appendix B).

Although this study focused on the proposed feature selection method in the context of LCS calibration, it is broadly applicable to other machine learning tasks that involve time series data. The flexibility of CCM, which can handle both linear and nonlinear dynamics, as well as deterministic and random data without specific assumptions, enhances the utility of the proposed feature selection approach. However, a key limitation is that CCM requires a sufficiently long time series to reliably determine causality, which may pose challenges in cases where data are not collected in continuous intervals.

## 5. Conclusions

In this study, we proposed a causality-based feature selection method using convergent cross mapping for the calibration of low-cost air quality sensor systems using machine learning. The integration of causality improved the interpretability and generalizability of the environmental machine learning models. The application of this approach to real-world low-cost sensor calibration demonstrated significant improvements in predictive performance and generalizability, confirming the efficacy of the proposed methodology.

In future work, we aim to validate this approach across various types of sensors and datasets to assess its robustness and adaptability in a range of applications in atmospheric and climate sciences. In particular, the mathematical rigor and versatility of convergent cross mapping, which underpins our feature selection method, could prove advantageous in empirical climate modeling applications.

## Figures and Tables

**Figure 1 sensors-24-07304-f001:**
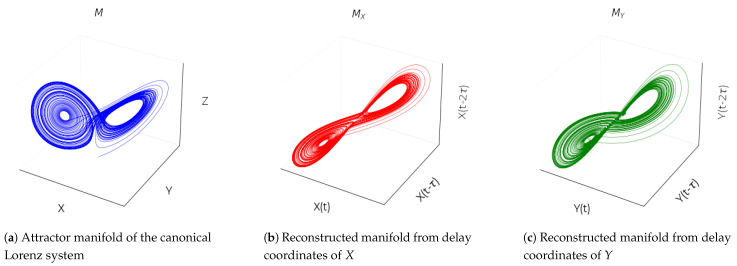
(**a**) Attractor manifold of the canonical Lorenz system (*M*) plotted in 3D space, showing the trajectory of the original system in the state space with variables *X*, *Y*, and *Z*. (**b**) Reconstructed manifold $M_{X}$ using delay-coordinate embedding of the *X* variable. The coordinates X(t), X(t−τ), and X(t−2τ) approximate the original attractor dynamics, capturing the structure of the system dynamics based only on the *X* time series. (**c**) Reconstructed manifold $M_{Y}$ using delay-coordinate embedding of the *Y* variable. The coordinates Y(t), T(t−τ), and Y(t−2τ) again form an attractor diffeomorphic to the original manifold, illustrating how the *Y* time series alone, through lagged coordinates, captures the dynamics of the system.

**Figure 2 sensors-24-07304-f002:**
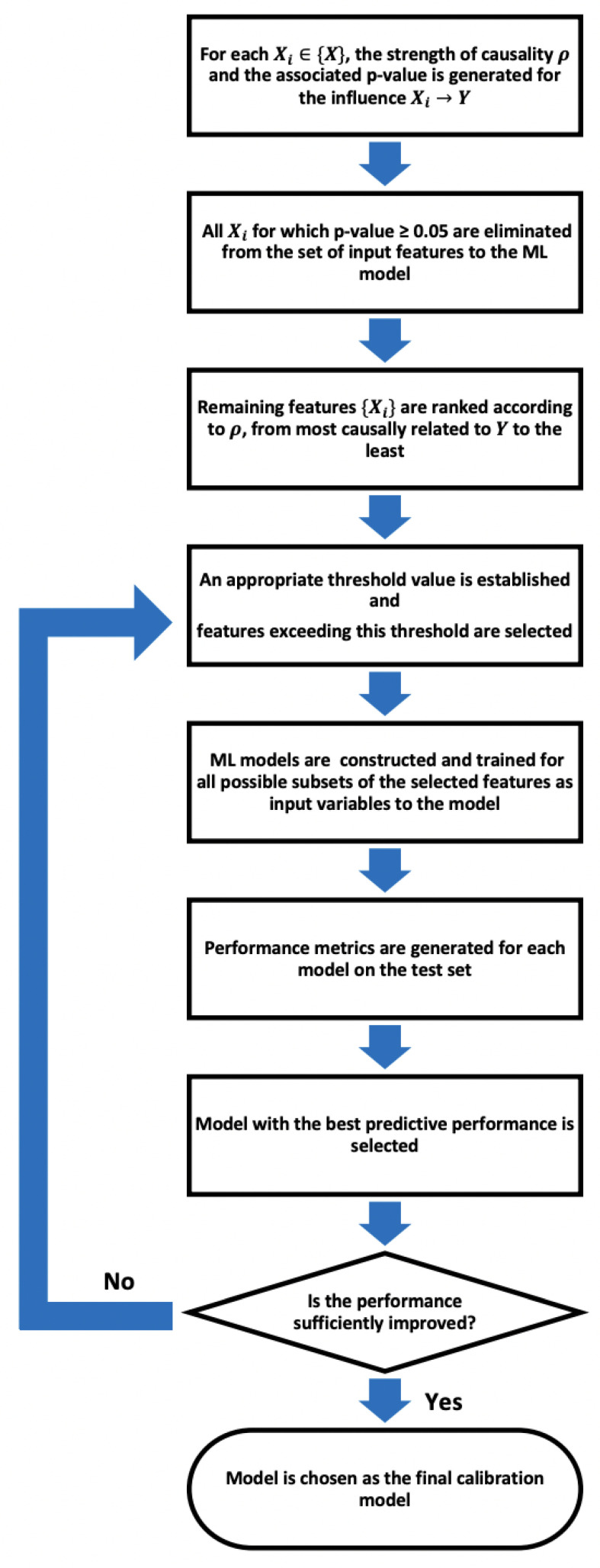
Proposed causality-driven feature selection pipeline.

**Figure 3 sensors-24-07304-f003:**
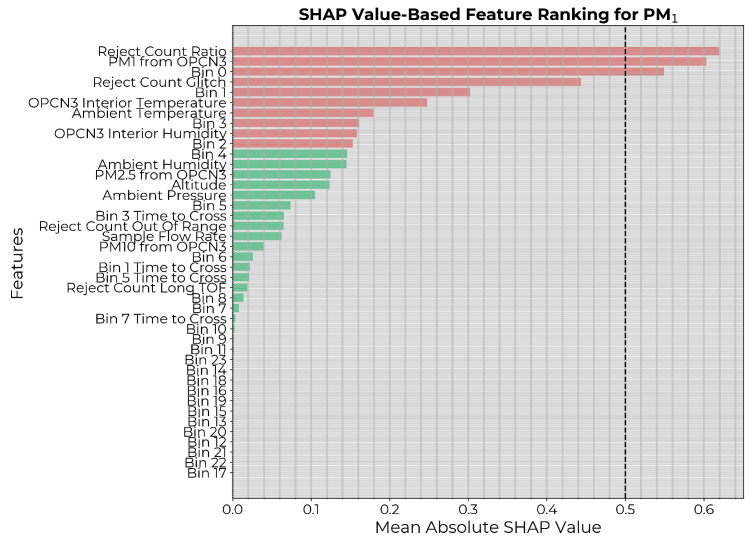
Input features to the ${\mathrm{PM}}_{1}$ calibration model ranked in descending order of mean absolute SHAP values. The 10 highest-ranked features are highlighted in red.

**Figure 4 sensors-24-07304-f004:**
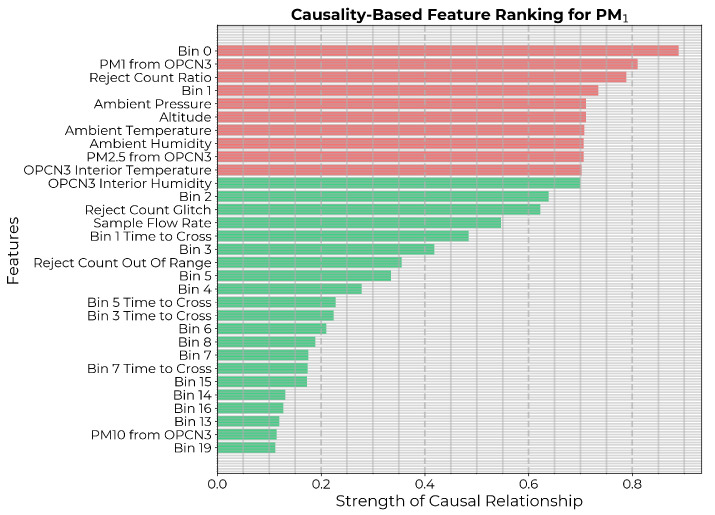
Potential input features to the ${\mathrm{PM}}_{1}$ calibration model ranked in descending order of strength of causal influence after eliminating features with *p*-value ≥0.05. The 10 highest-ranked features are highlighted in red.

**Figure 5 sensors-24-07304-f005:**
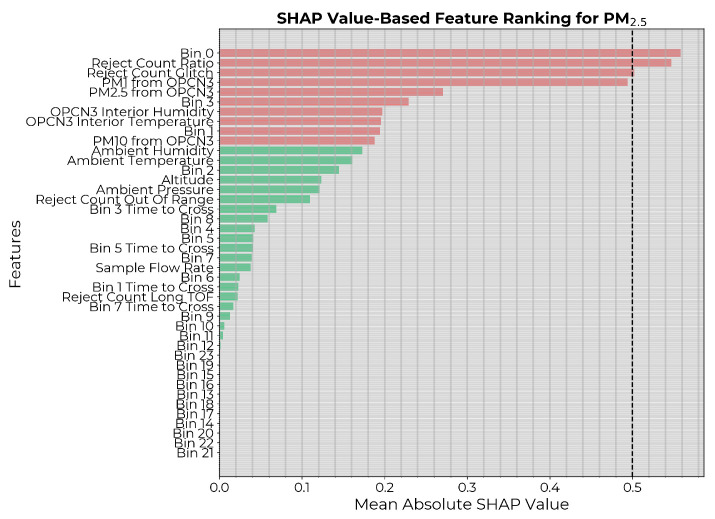
Input features to the ${\mathrm{PM}}_{2.5}$ calibration model ranked in descending order of mean absolute SHAP values. The 10 highest-ranked features are highlighted in red.

**Figure 6 sensors-24-07304-f006:**
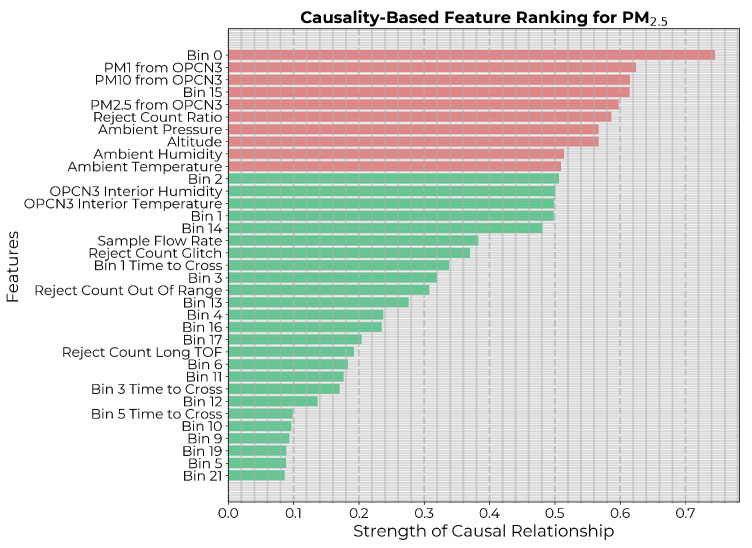
Potential input features to the ${\mathrm{PM}}_{2.5}$ calibration model ranked in descending order of strength of causal influence after eliminating features with *p*-value ≥0.05. The 10 highest-ranked features are highlighted in red.

**Figure 7 sensors-24-07304-f007:**
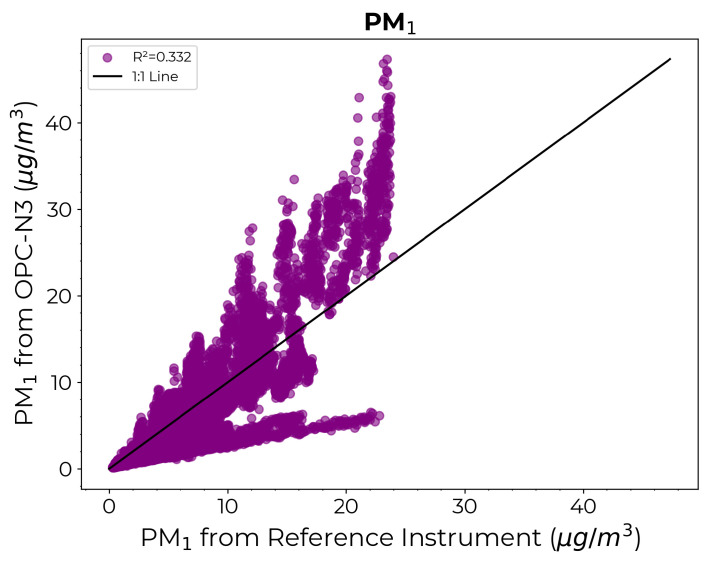
Scatter diagram comparing the ${\mathrm{PM}}_{1}$ measurements from the reference instrument on the x-axis against the ${\mathrm{PM}}_{1}$ estimates from OPC-N3 on the y-axis prior to calibration.

**Figure 8 sensors-24-07304-f008:**
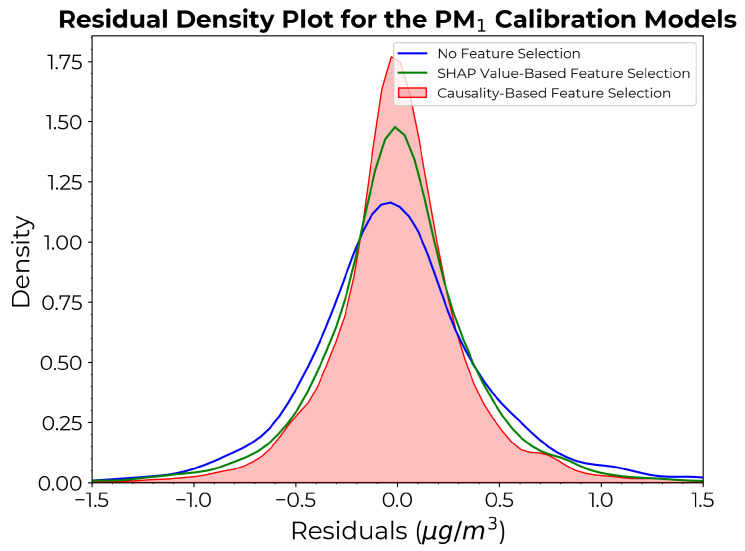
Density plots of the residuals for the ${\mathrm{PM}}_{1}$ calibration models derived from each approach.

**Figure 9 sensors-24-07304-f009:**
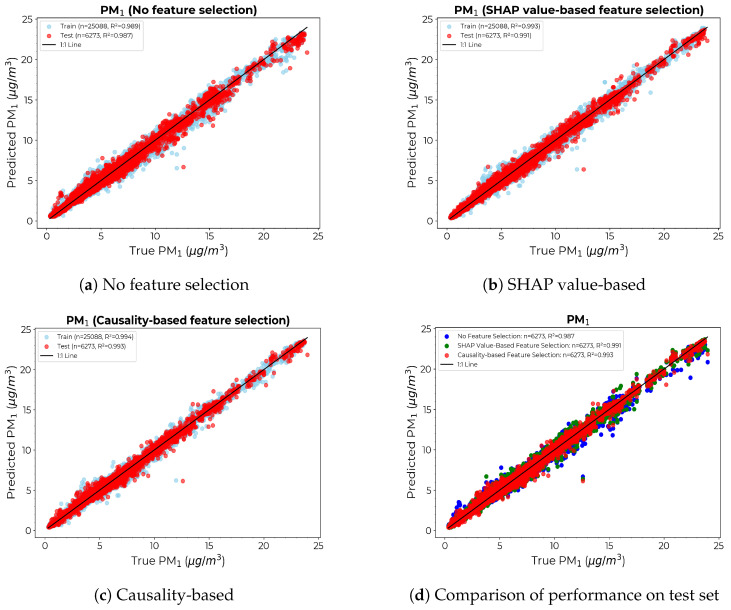
Scatter diagrams for the calibration models with the x-axis showing the ${\mathrm{PM}}_{1}$ count from the reference instrument and the y-axis showing the ${\mathrm{PM}}_{1}$ count provided by calibrating the LCS: (**a**) Without any feature selection. (**b**) SHAP value-based feature selection. (**c**) Causality-based feature selection. (**d**) Comparison of true vs. predicted values for the test set across models.

**Figure 10 sensors-24-07304-f010:**
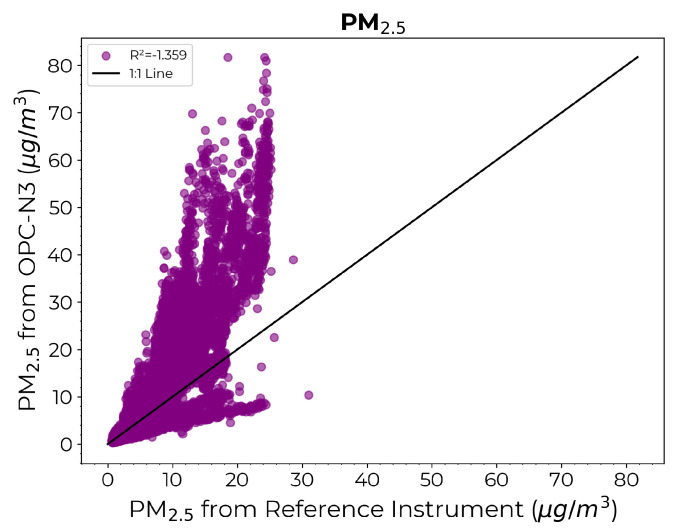
Scatter diagram comparing the ${\mathrm{PM}}_{2.5}$ measurements from the reference instrument on the x-axis against the ${\mathrm{PM}}_{2.5}$ estimates from OPC-N3 on the y-axis prior to calibration.

**Figure 11 sensors-24-07304-f011:**
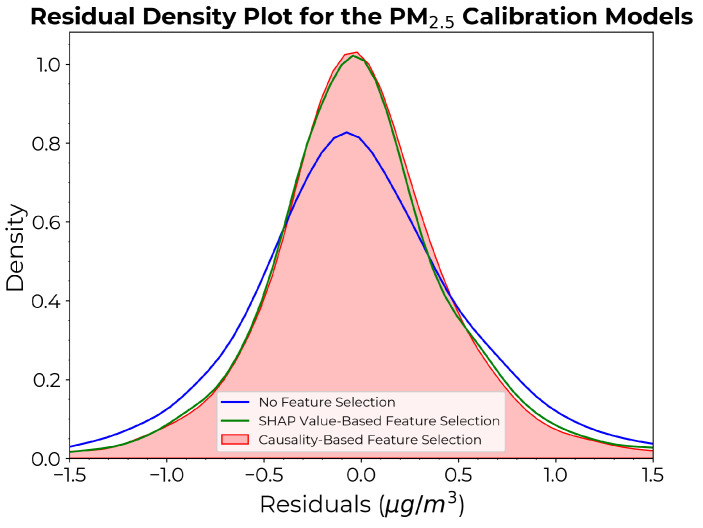
Density plots of the residuals for the ${\mathrm{PM}}_{2.5}$ calibration models derived from each approach.

**Figure 12 sensors-24-07304-f012:**
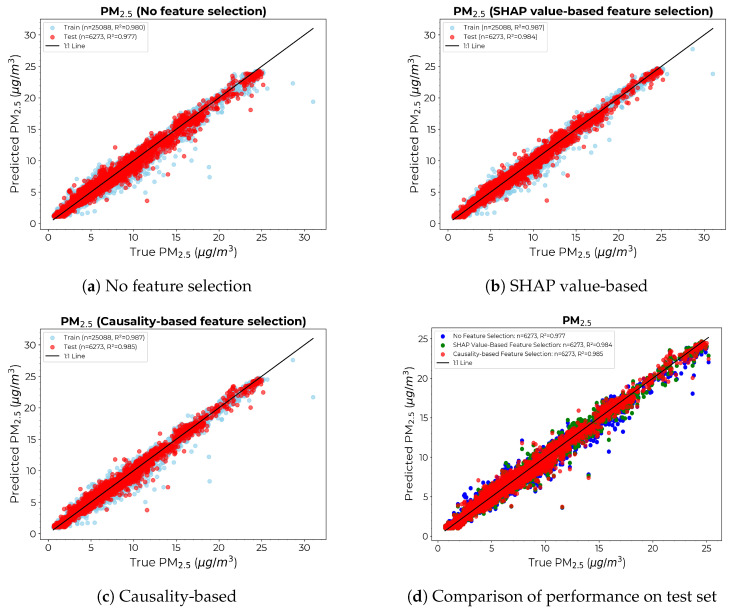
Scatter diagrams for the calibration models with the x-axis showing the ${\mathrm{PM}}_{2.5}$ count from the reference instrument and the y-axis showing the ${\mathrm{PM}}_{2.5}$ count provided by calibrating the LCS: (**a**) Without any feature selection. (**b**) SHAP value-based feature selection. (**c**) Causality-based feature selection. (**d**) Comparison of true vs. predicted values for the test set across models.

**Table 1 sensors-24-07304-t001:** The performance evaluation metrics for the estimation of ${\mathrm{PM}}_{1}$.

Feature Selection Approach	Features Used as Predictors	Number of Predictors	MSE	$R^{2}$
No feature selection	All 42 outputs from the LCS	42	0.213	0.987
SHAP value-based	Reject Count Ratio, PM1 from OPCN3, Reject Count Glitch, OPCN3 Interior Temperature, Ambient Temperature, OPCN3 Interior Humidity	6	0.150	0.991
Causality-based	Bin 0, Reject Count Ratio, Ambient Pressure, Ambient Temperature, Ambient Humidity	**5**	**0.121**	**0.993**

**Table 2 sensors-24-07304-t002:** The performance evaluation metrics for the estimation of ${\mathrm{PM}}_{2.5}$.

Feature Selection Approach	Features Used as Predictors	Number of Predictors	MSE	$R^{2}$
No feature selection	All 42 outputs from the LCS	42	0.41	0.977
SHAP value-based	Bin 0, Reject Count Ratio, Reject Count Glitch, Bin 3, PM1 from OPCN3, PM2.5 from OPCN3, OPCN3 Interior Temperature, OPCN3 Interior Humidity, Bin 1	9	0.286	0.984
Causality-based	Bin 0, PM1 from OPCN3, PM2.5 from OPCN3, Reject Count Ratio, Ambient Temperature, Ambient Pressure, Ambient Humidity	**7**	**0.274**	**0.985**

## Data Availability

The code and data are publicly available at https://github.com/mi3nts/Causality-Driven-Machine-Learning (accessed on 8 November 2024).

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
