# Peer review of "Causality-Driven Feature Selection for Calibrating Low-Cost Airborne Particulate Sensors Using Machine Learning"

_sensors, 2024, doi:10.3390/s24227304_

Round 1

Reviewer 1 Report

Comments and Suggestions for Authors

The authors employed a causality analysis method, convergent cross mapping (CCM), to select features for a low-cost air quality sensor calibration. This method was chosen for its ability to identify causal relationships between variables, which is crucial for sensor calibration. Several features were selected based on a metric computed from CCM, and the subsequent odor classification results validated the effectiveness of the proposed method. While the use of CCM for feature selection is a novel approach, the authors should address the following concerns:

  1. Section 2.1 needs to be rewritten. The current version uses text to illustrate the process of the proposed method, which is not enough considering the rigor of scientific expression. A mathematical description is also needed.   
  2. A more detailed explanation of the mathematical assumption and procedure of CCM would enhance the readability of the manuscript and ensure that the method is fully understood by the reader.
  3. Why can CCM be used in optical gas sensor calibrations? Please give the rationality in view of the sensing principle of an optical gas sensor.
  4. Information on the optical sensors and the feature generation method should be presented.
  5. Other causality analysis methods, such as Granger causality analysis, should be used in performance comparisons besides CCM. The results of these methods may help to solve the doubt in Comment 3.
  6. The presented results are like outputs of a classification instead of a calibration process. There is no comparison between with and without the calibration.
  7. The sensor type should be emphasized in the title, abstract, and introduction. Different odor-sensing principles result in signals with different characteristics, and different characteristics cannot be dealt with by a unique method.

Author Response

Thanks, we really appreciate your time! Please see attached file with our detailed replies to the reviewers comments.

Reviewer 2 Report

Comments and Suggestions for Authors

My general evaluation of the “Causality-Driven Feature Selection for Calibrating Low-Cost Air Quality Sensors Using Machine Learning” is as follows. This manuscript proposes a causality-driven feature selection for the machine learning regression model to realize low-cost air quality sensing methods. It is believed that the following corrections will be beneficial for the strengthening of the article.

1 – Was the overfitting issue considered enough during the Gradient Boosting Regressor training process used for prediction? This is important when comparing the accuracy.

2 – In the regression problems, the R-squared value above 0.99 is generally considered to be an effective model. For PM1, the SHAP-based and Causality-based models each showed performance improvements to 0.991 and 0.993, and for PM2, 0.984 and 0.985, respectively, with almost similar improvements. Can this be considered a statistically significant difference between the two methods? In addition, is there a possibility that the proposed method shows improved performance only in a particular split of the dataset? It is necessary to validate whether the proposed method remains effective in training and test datasets that are randomly shuffled and split.

3 – For the model training, was input and output data scaling applied? If so, an explanation of the scaling method used should be added to the manuscript. In addition, was MSE calculated using values that were inversely scaled? Furthermore, each plot requires units to be added on the respective axes.

4 – In this study, the authors used 10 input features determined by the Causality method. In addition, based on the number of input features, the same 10 features were selected by high SHAP as a comparison. Although the general threshold is 0.5, the high variance among scores may result in an appropriate comparison. A comparison with 4, 5, or 6 input features may also be necessary.

5 – The introduction may refer to additional research related to machine learning techniques for air quality prediction. Some suggested references are DOI: 10.1016/j.buildenv.2023.110191, DOI: 10.1016/j.chemosphere.2023.139518, DOI: doi.org/10.1016/j.envpol.2024.123371, DOI: 10.1038/s41598-023-28287-8

Author Response

(The authors gave the same response as above.)

Round 2

Reviewer 2 Report

Comments and Suggestions for Authors

The author answered my question adequately.